Polyandry contributes to Gonipterus platensis (Coleoptera: Curculionidae) rearing

http://orcid.org/0000-0003-1909-8709 Ribeiro Murilo Fonseca 1 2 murilo.fon.rib@gmail.com
Cavallini Gabriela 2
Solce Gabriel Negri 2
Favoreto Ana Laura 2
Passos José Raimundo De Souza 3
Hurley Brett 4
http://orcid.org/0000-0001-9875-4158 Wilcken Carlos Frederico 2
1 Instituto de Pesquisas e Estudos Florestais , Piracicaba, São Paulo , Brazil
2 Departamento de Proteção Vegetal/Faculdade de Ciências Agronômicas, Universidade Estadual Paulista , Botucatu, São Paulo , Brazil
3 Departamento de Biodiversidade e Bioestatística/Instituto de Biociências, Universidade Estadual Paulista , Botucatu, São Paulo , Brazil
4 Department of Zoology and Entomology/Forestry and Agricultural Biotechnology Institute (FABI), University of Pretoria , Pretoria, Gauteng , South Africa
Brygadyrenko Viktor
Electronic publication date: 2024 Aug 22
Publication date: 2024
Volume: 12
Electronic Location ID: e17929
Received 2024 Feb 8; Accepted 2024 Jul 25
Copyright: © 2024 Ribeiro et al.
Copyright year: 2024
Copyright holder: Ribeiro et al.
License: This is an open access article distributed under the terms of the Creative Commons Attribution License, which permits unrestricted use, distribution, reproduction and adaptation in any medium and for any purpose provided that it is properly attributed. For attribution, the original author(s), title, publication source (PeerJ) and either DOI or URL of the article must be cited.
License URL: https://creativecommons.org/licenses/by/4.0/

Keywords: Eucalyptus snout-beetle, Fecundity, Fertility, Male harassment, Monogamy, Biological control, Forest entomology, Polyandry

Funding: Conselho Nacional de Desenvolvimento Cientifico e Tecnológico (CNPq) Coordenação de Aperfeiçoamento de Pessoal de Nível Superior (CAPES) Finance Code 001 Programa Cooperativo sobre Proteção Florestal (PROTEF) Instituto de Pesquisas e Estudos Florestais (IPEF) This study was funded by the Conselho Nacional de Desenvolvimento Cientifico e Tecnológico (CNPq), the Coordenação de Aperfeiçoamento de Pessoal de Nível Superior (CAPES)-Finance Code 001, and the Programa Cooperativo sobre Proteção Florestal (PROTEF) of the Instituto de Pesquisas e Estudos Florestais (IPEF). The funders had no role in study design, data collection and analysis, decision to publish, or preparation of the manuscript.

==============================
Background

Gonipterus platensis Marelli, 1926 (Coleoptera: Curculionidae) is one of the main defoliating beetles in Eucalyptus plantations. Biological control with egg parasitoids is one of the main control strategies for this pest and a constant supply of fresh host eggs is required to rear the parasitoids. Polyandry can influence Gonipterus oviposition by increasing female fecundity and fertility; however, the high density of individuals in laboratory colonies can lead to male harassment, resulting in lower reproduction rate. The aim of this study was to measure the effects of monoandry and polyandry on the reproduction of G. platensis and the effects of male harassment on laboratory rearing conditions.

Methods

Reproductive parameters were compared between three treatments: monoandry, where the female was allowed to mate daily with the same male; no choice polyandry, where the female was allowed to mate daily with a different male; and polyandry with choice, where the female was allowed to mate daily, but with a choice between five different males. Another experiment varying the density of males was conducted to evaluate the effect of male harassment.

Results

Polyandry with choice resulted in the longest period of oviposition, highest fecundity and highest number of eggs per egg capsules when compared to monoandrous females. No negative effect related to male harassment in the laboratory, such as decreased fertility, fecundity, or number of eggs per egg capsule, was detected.

Conclusion

Polyandry contributes to mass rearing as it increases fecundity and oviposition period on females and there is no evidence of male harassment on G. platensis.

Introduction

Gonipterus platensis Marelli, 1926 (Coleoptera: Curculionidae), also known as the Eucalyptus snout-beetle, is native to Australia but invasive in several countries, and is considered one of the main defoliating beetles in Eucalyptus plantations worldwide (Mapondera et al., 2012; Hurley et al., 2016; Schröder et al., 2020). The distribution of the insect is also expanding, as G. platensis was recently introduced in Ecuador (Crespo-Pérez et al., 2023) and detected in new regions in Brazil (Ribeiro et al., 2023). Infestations of G. platensis can result in considerable productivity loss, such as a 42.8% reduction in the Mean Annual Increment (MAI) in Eucalyptus grandis x Eucalyptus dunnii hybrids, reported in Brazil (Souza et al., 2016), and wood loss of more than 85% in Eucalytptus globulus Labill, reported in Portugal (Reis et al., 2012). Damage is mainly caused by the larval phase of these insects.

The first control strategy for Gonipterus spp. occurred with the introduction of the egg parasitoid Anaphes nitens Girault, 1928 (Hymenoptera: Mymaridae) in South Africa in 1926 (Tooke, 1955). The success of this control in South Africa led to the introduction of this parasitoid in other countries, including in South America and Europe, for the control of G. platensis (at that stage thought to be the same species as present in South Africa–see Mapondera et al., 2012). Currently, biological control is considered one of the main control strategies for Gonipterus spp. (Schröder et al., 2020). However, recent outbreaks have motivated the search for new natural enemies in the region of origin of these insects, with emphasis on other species of egg parasitoids such as Anaphes inexpectatus Huber & Prinsloo, 1990 and Centrodora damoni Girault, 1922 (Hymenoptera: Aphelinidae) (Garcia et al., 2019; Schröder et al., 2021; Souza, Lawson & Nahrung, 2021).

When natural enemies are imported as part of a Classical Biological Control (CBC) programme, they are generally required to remain in a certified quarantine facility until host-specificity and other tests have been completed and permission has been granted for field releases (Valente et al., 2017; Kenis et al., 2019). During this period, the natural enemies are often reared on the target insect, and thus an adequate understanding of the biology of the host is required to establish a healthy colony of the pest and the natural enemy. In the case of importing new parasitoids for the control of G. platensis, an established laboratory colony producing fresh eggs of G. platensis is fundamental for the multiplication of these parasitoids. However, the factors that influence Gonipterus oviposition are not fully understood and long periods without oviposition are common.

Polyandry, namely when a female mates with different males, has been reported for G. platensis. Adults of this insect have an average mating period of 7 h, reaching a maximum of 55 h (Santolamazza-Carbone & Cordero-Rivera, 1998), but the female does not lay eggs directly after mating, being free to mate again with different males. Polyandry is commonly found in insects, but its effects on females are complex and can lead to increased risks of predation, infection, or physical injuries by males (Arnqvist & Nilsson, 2000; Crudgington & Siva-Jothy, 2000). To balance the possible negative effects, reproductive gains must occur in return, commonly related to increase in female fecundity and fertility, faster development and increase on progeny survival (Bayoumy & Michaud, 2014; Fan et al., 2015). Polyandry can also lead to the production of more genetically diverse progeny when compared to females that mated only once, potentially increasing the ability of the new generation to adapt to and establish in a new location (Yasui, 1998; Rafter et al., 2018; Lewis et al., 2020).

These interactions may vary according to the insect group under analyses. For Lepidoptera, there is an increase in fecundity for species wherein polyandrous behavior is more common than monandrous (Torres-Vila, Rodríguez-Molina & Jennions, 2004). For Ephestia kuehniella Zeller, 1879 (Lepidoptera: Pyralidae), heavier moths, which have higher fecundity and fertility, are more likely to remate and can also replace most of the sperm received from a previous mate if they mate with a higher quality male afterwards (Xu & Wang, 2020). For Pentatomidae, results of polyandry can be variable. Nezara viridula (Linnaeus, 1758) fecundity decreases in females that mate with several males and no impact on fertility was observed, while for Dichelops furcatus (F., 1775) the opposite occurs, with an increase in fertility in multi-copulated females. In both species, females that copulated few times concentrate egg laying activity in the first half of their reproductive period, while females with multiple mates lay eggs throughout the lifespan (Fortes & Consoli, 2011; Cingolani et al., 2020). In a mass rearing system designed to produce eggs for biological control programs it may be advantageous to remove males after the firsts matings so that the egg production is concentrated over a shorter period of the insect lifespan (Fortes & Consoli, 2011).

The high density of individuals in laboratory colonies allows promiscuous behavior, with multiple mates by males and females. In nature, females have the option of avoiding areas with a high density of males, but this is not possible in the laboratory. Thus, in laboratory conditions, the high number of males can lead to male harassment, a sexual coercion where males repeatedly try to copulate with unreceptive females (Hollander & Gwynne, 2009), resulting in lower female longevity, reproduction rate, number of eggs per oviposition, and changes in behavior, e.g., high male densities at the oviposition site may induce females to change their oviposition preferences (Gay et al., 2009; Bacon & Barbosa, 2020). In these situations, the cost related to multiple copulations and/or attempts can outweigh the benefits. To better understand the dynamics of polyandry in the laboratory, it is necessary to assess possible negative aspects of male harassment and the costs and benefits of multiple copulations (Hollander & Gwynne, 2009).

The aim of this study was to measure the effects of monoandry and polyandry on the reproduction of G. platensis and the effects of the constant presence of males on laboratory rearing conditions for biological control programs. Our hypotheses were that (1) polyandry stimulates female fertility and fecundity (2) cohabitating with males, and thus male harassment, decreases female fecundity and the number of eggs per egg capsule.

Materials and Methods

Gonipterus platensis rearing

Gonipterus platensis adults were reared in wooden cages (40 cm × 45 cm × 80 cm) with a glass top and voile fabric sides. Each cage contained 25 couples, fed with tender leaves and shoots of Eucalyptus urophylla S. T. Blake, 1977, forming a bouquet of branches. The bouquets were placed in 500 mL plastic containers with water within the cages. The leaves also served as an oviposition substrate. The cages were kept in an environment-controlled room, with temperature at 25 ± 1°C, relative humidity at 50 ± 10% and a photoperiod of 12:12 h.

The egg capsules collected from the rearing were kept in 5 cm diameter acrylic plates in the same room. The hatching of the larvae was evaluated daily, and they were transferred to a specific bouquet for feeding, where they remained until the pre-pupa stage, when they stop feeding and move towards the bottom of the cage. For pupation, 10 pre-pupae were placed in 1 L plastic containers with 150 g of autoclaved sand moistened with 15 mL of distilled water, until adult emergence.

Polyandry

To measure the effect of polyandry, newly emerged insects were separated according to sex in wooden cages following the rearing protocols. When the insects were 1 month old, they were confined in couples in one liter plastic containers (11 cm × 14 cm × 9 cm) with a branch of E. urophylla, one of the best hosts for G. platensis (Oliveira et al., 2022), and divided according to the following treatments: (1) female was allowed to mate daily with the same male (monoandry), (2) female was allowed to mate daily with a different male (no choice polyandry), (3) female was allowed to mate daily, being able to choose among five different males (polyandry with choice). Each treatment had ten replicates. The insects were placed in each container according to treatment every day at 8:00 am and separated at the end of the day at 6:00 pm. The occurrence of mates was checked every 30 min, this was allowed due to the long period of mating on G. platensis, with minimum of 0.7 h (42 min) (Santolamazza-Carbone & Cordero-Rivera, 1998). Females were kept isolated after the first oviposition. There was no differentiation between virgin and non-virgin males, as there is no difference between mate duration or volume of sperm by males (Santolamazza-Carbone & Cordero-Rivera, 1998) (Fig. 1).

Figure 1 Graphic abstract for polyandry experiment.

One-month-old Gonipterus platensis adults were confined daily in a plastic container to stimulate mating, based on the following treatments: Monogamy, where female was allowed to mate daily with the same male; polyandry no choice, where female was allowed to mate daily with a different male; polyandry with choice, where female could select from five males daily. Males were replaced between 8 am and 6 pm daily until the first oviposition. Subsequently, females were isolated, and the containers were checked daily for the presence of egg capsules. Photos credit: Murilo F. Ribeiro. Ilustration: Microsoft PowerPoint.

The branches were replaced every 3 days, serving both as food and oviposition substrate. The oviposited egg capsules were collected daily, and the eggs were kept in a biochemical oxygen demand (B.O.D.) chamber at 25 ± 1 °C; RH 60% and photoperiod 12:12 h until hatching. Five days after the end of the last hatching, the egg capsules were dissected using a Nikon SMZ645 stereomicroscope to count infertile eggs and hatched larvae.

The number of mates until the first oviposition, the pre-oviposition period (period from the first mating to the first oviposition); oviposition period (period between the first and last oviposition); total number of eggs; percentage of infertile eggs and number of eggs per egg capsule were evaluated for 7 months.

To measure the effect on progeny we analyzed 10 larvae of each female from treatments 1 to 3, each larva was kept in a 500 mL plastic container (9.5 cm × 11 cm × 7.5 cm) and fed with E. urophylla until it reached the pre-pupal stage. Larval development time, mortality and pre-pupal mass were evaluated.

Male harassment

To measure the effects of male harassment, newly emerged G. platensis were sexed and separated following the same procedures as above and later divided according to the treatments, using virgin insects: (H1) female mated once and kept isolated after mating; (H2) female stayed together with the same male during reproductive period; (H3) female stayed together with two males during reproductive period. Each treatment had 10 replicates and were evaluated for 11 months (Fig. 2).

Figure 2 Graphic abstract for the male harassment experiment.

One-month-old Gonipterus platensis adults were confined daily in a plastic container to stimulate mating, based on the following treatments: H1, where female mated once; H2, where female and male remained together throughout the reproductive period; H3, where the female remained with two males. Individuals in treatments H2 and H3 were allowed to mate during the entire reproductive period. The containers were checked daily for the presence of egg capsules. Photos credit: Murilo F. Ribeiro. Ilustration: Microsoft PowerPoint.

The methodology used to collect eggs and measure the effect on progeny followed the same procedures used for polyandry experiments.

Statistical analyses

The statistical analyzes of both experiments were adjusted according to generalized linear models (Nelder & Wedderburn, 1972) and varied according to the distribution and linkage function (Table 1).

Table 1 Factors, distribution and linkage function used in the statistical analysis of each of the variables measured for the polyandry and male harassment experiments.

Experiment	Response variable	Distribution	Linkage function	
Polyandry	Pre-oviposition; oviposition; Eggs/egg capsule	Normal	Identity	
Number of mates	Poisson	Logarithmic	
Total of eggs	Negative binomial	Logarithmic	
Percentage of infertile eggs	Binomial	Logit	
Polyandry progeny	Pre-pupa mass	Normal	Identity	
Development time	Gamma	Logarithmic	
Male harassment	Pre-oviposition; Number of eggs per day	Normal	Identity	
Oviposition; Percentage of infertile eggs; Eggs/egg capsule	Gamma	Logarithmic	
Total of eggs	Negative binomial	Logarithmic	
Male harassment progeny	Development time; pre-pupae mass	Gamma	Logarithmic	

The quality of the residues was fitted in all models through the analysis of deviations (deviance), standardized Pearson residuals plots. For comparisons between treatments, the Tukey-Kramer test (Westfall et al., 1999) of the genmod procedure of the SAS statistical program–Free Statistical Statistical Software, SAS University Edition, was used.

Results

Polyandry

Results were analyzed only for females that laid eggs during the evaluation period. Treatment had a significant effect on the number of matings until the first oviposition (p = 0.0021) and the oviposition period (p = 0.0312), but not on the pre-oviposition period (p = 0.7162, Fig. 3).

Figure 3 Pre-oviposition and oviposition period (days) of Gonipterus platensis for monoandry, no choice polyandry and polyandry with choice treatments.

There were a significantly higher number of matings for females from the polyandry with choice treatment compared to females from the no choice polyandry treatment. The number of matings from females in the monoandry treatment was the second highest, but not significantly different from either of the other treatments (Fig. 4).

Figure 4 Number of matings, number of eggs, percentage of infertile eggs and number of eggs per egg capsule of Gonipterus platensis for monoandry, no choice polyandry and polyandry with choice treatments.

The oviposition period was significantly higher for females from the polyandry with choice treatment compared to females from the monoandry treatment (which has the shortest oviposition period), but not significantly higher than females from the no choice polyandry treatment, which had an intermediate oviposition period (Fig. 4). (Tables are available in Supplemental Material).

Treatment had a significant effect on the total number of eggs (p = 0.0196) and the number of eggs/egg capsule (p = 0.0095), but not on the percentage of infertile eggs (p = 0.8289, Fig. 4). The total number of eggs was highest in the treatment of polyandry with choice, which was significantly higher than for the monoandry treatment, but not for the no choice polyandry treatment, which had an intermediate number of eggs. The number of eggs/egg capsule was significantly lower in the monoandry treatment as compared to polyandry with choice and no choice polyandry (Fig. 4).

Treatment did not have a significant effect on larval development time (p = 0.9139) or pre-pupal mass (p = 0.1303, Table 2). The number of larvae evaluated in the progeny varied between 54–64 and the mortality varied from 35–40% (Table 2).

Table 2 Total number of larvae (N), larval development period (days), mortality (%) and pre-pupa mass (g) (mean ± SE) of Gonipterus platensis progeny on monogamy, no choice polyandry and polyandry with choice.

Treatment	N	Mortality (%)	Larval development time (days)	Pre-pupae mass (g)	
Monogamy	54	35.19	19.41 ± 0.37 a	0.109 ± 0.006 a	
Polyandry no choice	64	39.06	19.28 ± 0.36 a	0.121 ± 0.006 a	
Polyandry with choice	90	40.00	19.19 ± 0.39 a	0.118 ± 0.004 a	
Note:

Means followed by the same lowercase letter on the column did not differ by the Tukey-Kramer test (p < 0.05).

Male harassment

Results were analyzed only for females that laid eggs during the evaluation period. The treatment, number of males during oviposition period, did not have a significant effect on pre-oviposition period (p = 0.2365), oviposition period (p = 0.4979), the total number of eggs (p = 0.9089) or the number of eggs per day (p = 0.0768) (Table 3). Treatment did have a significant effect on the percentage of infertile eggs (p < 0.0001), which was significantly higher in H1, and on the number of eggs per egg capsule (p = 0.0292), which was higher in H3 (Table 3).

Table 3 Pre-oviposition and oviposition period (days), total number of eggs, infertile eggs (%), eggs per day and eggs/egg capsule (mean ± SE) of Gonipterus platensis females on different male densities.

Dependent variable	Number of males during oviposition period	
H1	H2	H3	
Pre-oviposition (d)	57.71 ± 47.32 a	28.4 ± 23.14 a	24.9 ± 22.17 a	
Oviposition (d)	216.57 ± 79.18 a	204.10 ± 83.28 a	135.70 ± 106.26 a	
Total of eggs	370.00 ± 165.58 a	317.00 ± 82.99 a	297.60 ± 96.58 a	
Infertile eggs (%)	44.25 ± 41.01 a	8.58 ± 6.60 b	4.31 ± 3.43 b	
Eggs/day	1.57 ± 1.36 a	1.66 ± 1.03 a	3.26 ± 2.30 a	
Eggs/egg capsule	4.18 ± 1.20 b	4.79 ± 0.68 ab	5.60 ± 1.17 a	
Notes:

Means followed by the same letter on the line did not differ by the Tukey-Kramer test (p < 0.05).

H1: female mated once and kept isolated after mating; H2: female stayed together with the same male during reproductive period; and H3: female stayed together with two males during reproductive period.

The treatment did not have a significant effect on the total development time of the larvae (p = 0.8018) and the pre-pupal mass (p = 0.4965) (Table 4). The number of larvae evaluated in the progeny varied from 50–96 and the percent mortality from 28–32%, according to the oviposition of females in each previous treatment.

Table 4 Total number of larvae (N), total larval development period (days), mortality (%) and pre-pupa mass (g) (mean ± SE) of Gonipterus platensis progeny on different parental male densities.

Treatment	N	Mortality (%)	Larval development time (days)	Mass (g)	
H1	50	30.00	17.67 ± 0.32 a	0.128 ± 0.003 a	
H2	96	32.29	17.68 ± 0.29 a	0.123 ± 0.003 a	
H3	88	28.41	18.00 ± 0.67 a	0.123 ± 0.003 a	
Notes:

Means followed by the same letter on the column did not differ by the Tukey-Kramer test (p < 0.05).

H1: female mated once and kept isolated after mating; H2: female stayed together with the same male during reproductive period; and H3: female stayed together with two males during reproductive period.

Discussion

The study indicated that polyandry with choice is a beneficial mating strategy for G. platensis and thus advantageous to facilitate mating strategy in rearing colonies. In agreement with the initial hypothesis, polyandry with choice resulted in the longest period of oviposition, highest fecundity and number of eggs per egg capsule when compared to monoandrous females. The polyandry with no choice treatment didn’t differ from monoandrous or polyandry with choice treatments in almost all parameters, presenting intermediate values among those treatments. In contrast to the initial hypothesis, there were no indications of male harassment in rearing conditions. Therefore, G. platensis rearing is favored by the constant presence of males, allowing multiple mates and with no negative interference over the female.

The number of matings until the first oviposition was higher in the polyandry with choice treatment but did not differ from the monoandry treatment. As mating can have gonadotrophic effects, increasing female fecundity (Alonzo & Pizzari, 2013), and mate duration in G. platensis does not influence the volume of seminal fluid ejaculated (Santolamazza-Carbone & Cordero-Rivera, 1998), it would be expected that similar fecundity values would be found in both treatments, since the females would receive similar volumes of sperm and seminal fluids. However, the oviposition period was longer and the number of eggs (fecundity) higher for the polyandry treatment and statistically different from monoandry. One possible explanation of this can be by the existence of pseudo-copulations in G. platensis, where the male even introduces the aedeagus, but the amount of sperm transferred is very low and unable to inseminate the female (Santolamazza-Carbone & Cordero-Rivera, 1998). If the same pattern also occurs with the seminal fluid, mating events may not be sufficient to stimulate the female fecundity and hence not a good parameter to measure fecundity in G. platensis. For E. kuehniella, when the first mating is with a low-quality male a smaller seminal fluid is delivered which increases the tendency for the female to remate, ensuring the polyandrous behavior (Xu & Wang, 2020).

The pre-oviposition period was not affected by the treatments. The seminal fluid provided by the male can accelerate ovulation and oviposition (Gillott, 2003), which benefits the female as it allows the production of eggs only when there is enough sperm for fertilization (Avila et al., 2011). In G. platensis, the lack of difference on pre-oviposition period between the monoandry and polyandry treatments indicate that the initial presence of seminal fluid is a sufficient stimulus to initiate oviposition, regardless of whether its origin is monoandrous or polygamous. On the other hand, the duration of the oviposition period was longer in the polyandric treatments, with the longest period found in polyandry with choice, which also had the highest fecundity (number of eggs) and number of eggs per egg capsule when compared to monoandrous females. The increase in polyandry-related fecundity may be attributed to an increase in sperm suitable for fertilization and/or nutrients and hormonal stimulants provided by seminal fluid by multiple males (Arnqvist & Nilsson, 2000; Omkar & Pandey, 2010). Evidence of increased fecundity associated with polyandry can be found in other beetles such as Chrysochus cobaltinus LeConte, 1857 (Coleoptera: Chrysomelidae) (Schwartz & Peterson, 2006), Cheilomenes sexmaculata Fabricius, 1781 (Coleoptera: Coccinelidae), Coccinella septempuctata L. 1758 (Coleoptera: Coccinelidae) and Propylea dissecta Mulsant, 1850 (Coleoptera: Coccinelidae) (Omkar & Mishra, 2005).

The fertility of G. platensis was not affected by the treatments. Fertile eggs were oviposited 218 days after the last mating on polyandry with choice treatment. This condition was allowed due to the presence of spermatheca in the female reproductive system, which in G. platensis can store viable sperm throughout life (Santolamazza-Carbone & Cordero-Rivera, 1998). A similar condition was observed for N. viridula, where polyandry females oviposited viable eggs throughout their reproductive period but no effect on fertility was observed when compared to females that mated once or twice (Fortes & Consoli, 2011). However, in highly promiscuous insects such as Hippodamia convergens Guerin-Meneville 1824 (Coleoptera: Coccinellidae) polyandry can restore female fertility, especially when they mate with non-sibling males (Bayoumy, Michaud & Bain, 2015).

Progeny characteristics were not influenced by the treatments of monoandry or polyandry. One of the possible positive aspects that can lead a species to polyandrous behavior is the gain of indirect benefits for the progeny, manifested in a better genetic ability to adapt to new environments and greater fitness, which can be inferred from the hatching rate, larval development, and body size (Yasui, 1998; Bayoumy & Michaud, 2014; Rafter et al., 2018). Pre-pupal mass and larval development time were used to analyze G. platensis progeny, but neither differed significantly between treatments. Theoretically, copulation with different males increases competition between sperm within the spermatheca and biases fertilization towards the group of male gametes with higher fitness (Bayoumy & Michaud, 2014). Females of G. platensis when mated with different males fertilize their eggs randomly (Santolamazza-Carbone & Cordero-Rivera, 1998) and relative gains in progeny could not be identified. The progeny characteristics were compared in the same environment and therefore it is not possible to say whether the genetic diversity of the polyandric treatments can influence the adaptation when subjected to adverse conditions, as observed in Zygogramma bicolorata Pallister, 1953 (Coleoptera: Chrysomelidae), where the progeny of polyandrous females showed better adaptation to higher temperatures (Omkar & Pandey, 2010).

When analyzing the reproductive parameters for situations in which females are exposed to different male proportions, it was not possible to detect negative influences related to male harassment in the laboratory, such as decreased fertility, fecundity, or number of eggs per egg capsule (Gay et al., 2009; Bayoumy & Michaud, 2014). The parameters related to the larval development time of the progeny also showed no differences for females exposed to several males. Male harassment can hinder insect rearing as observed for Anastrepha ludens Loew 1873 (Diptera: Tephritidae) and Aedes albopictus Skuse, 1984 (Diptera: Culicidae), where lower male densities in the colony result in higher production quality (Zhang et al., 2018; González-López et al., 2019). For the monandrous insect Aedes aegypti L. 1762 (Diptera: Culicidae), male harassment has an impact on rearing and in the field, by reducing female longevity through feeding inhibition (Helinski & Harrington, 2012; Zhang et al., 2024). It is not possible to confirm that G. platensis female longevity was negatively affected by male density, as it was not analyzed.The percentage of infertile eggs was significantly higher in the treatment in which females remained isolated after mating. Theoretically, enough sperm can be transferred through a single mate, ensuring reproductive success (Arnqvist & Nilsson, 2000). The high percentage of infertile eggs (44.25 ± 41.01%) in the treatment in which the females were isolated after a single mating and the fact that three of these did not even start oviposition indicates that males of G. platensis do not provide enough sperm to fertilize all eggs in a single mating, further supporting the importance of remating for G. platensis. Low fertility negatively impacts the establishment of the base population in the laboratory colony. In addition, egg parasitoids are capable of differentiating between fertile and infertile eggs, the latter having lower rates of parasitism and longer development times and failures of emergence of parasitoid progeny (Krugner, 2014; Du et al., 2018), hence infertile eggs are not beneficial for biological control rearing programs.

The number of eggs per egg capsule was influenced by the proportion of G. platensis males during the female’s reproductive period. Females that remained with two males oviposited larger egg capsules than those that were isolated after mating. This result is different from that reported for situations in which male harassment occurs, as observed in Callosobruchus maculatus Fabricius, 1755 (Coleoptera: Chrysomelidae: Bruchinae), where male harassment interferes on the oviposition substrate and induces lower postures due to energy costs and time for females to flee from males (Bacon & Barbosa, 2020).

Conclusion

Polyandry contributes to G. platensis rearing as it increases fecundity and oviposition period. The increase of number of eggs assists biological control programs that rely on laboratory reared insects, especially egg parasitoids. There is no evidence of negative characteristics for G. platensis egg production or progeny in laboratory rearing associated with male harassment.

Supplemental Information

Supplemental Information 1 Graphic abstract.

The abstract indicates that in a experiment for Gonipterus platensis rearing the Polyandry with choice treatment resulted in the longest period of oviposition, highest fecundity and highest number of eggs per egg capsules when compared to monoandrous females. The polyandry with no choice treatment didn’t differ from monoandrous or polyandry with choice treatments in almost all parameters, presenting intermediate values among those treatments. The pre-oviposition period, fertility and effect of the treatments on progeny were not significantly affected.

Supplemental Information 2 Number of matings per female, pre-oviposition, oviposition period, number of eggs, percentage of infertile eggs and number of eggs per egg capsule for Gonipterus platensis with Mean ± SE.

Supplemental Information 3 Raw data: Polyandry/Monogamy experiments.

Supplemental Information 4 Raw data: Male Harassment experiments.

Additional Information and Declarations

Competing Interests

Author Contributions

Data Availability

The authors declare that they have no competing interests.

Murilo Fonseca Ribeiro conceived and designed the experiments, performed the experiments, analyzed the data, prepared figures and/or tables, authored or reviewed drafts of the article, and approved the final draft.

Gabriela Cavallini performed the experiments, authored or reviewed drafts of the article, and approved the final draft.

Gabriel Negri Solce performed the experiments, authored or reviewed drafts of the article, and approved the final draft.

Ana Laura Favoreto performed the experiments, authored or reviewed drafts of the article, and approved the final draft.

José Raimundo De Souza Passos analyzed the data, prepared figures and/or tables, authored or reviewed drafts of the article, and approved the final draft.

Brett Hurley analyzed the data, prepared figures and/or tables, authored or reviewed drafts of the article, and approved the final draft.

Carlos Frederico Wilcken conceived and designed the experiments, analyzed the data, prepared figures and/or tables, authored or reviewed drafts of the article, and approved the final draft.

The following information was supplied regarding data availability:

The raw measurements are available in the Supplemental Files.

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
