# Peer review of "Polyandry contributes to Gonipterus platensis (Coleoptera: Curculionidae) rearing"

_PeerJ, doi:10.7717/peerj.17929_

## Round 0.1 · original submission · Major Revisions

The manuscript represents what appears to be a very carefully conducted study that required a significant investment of time.

The results are presented quite correctly. The authors clearly identified independent and dependent variables.

However, the introduction and discussion do not contain a detailed analysis of the biological phenomenon studied by the authors. The authors mainly limited themselves to the literature on a particular species of beetle. The broader context needs to be analyzed.

The main drawback of the manuscript is that the Materials and Methods are described very briefly. After reading this section, the reader should have an understanding of the thoroughness of control over all factors that could influence the results. The reviewers and I had justifiable doubts about the possible influence of numerous uncontrollable factors on the results. Was there video recording? Perhaps the article will be much better if photographic materials are included in it. For example, one large drawing (full page), which will present 12-16 photographs of individual stages of the experiment in Materials and Methods. Probably we need another drawing with 12-16 photos in the Results section. The figure in Materials and Methods should illustrate the independent factors that influenced the system under study. The picture in the Results should reflect the results of the impact, for example, photographs of different experimental results.

The article will also attract significantly more readers if in the graphical abstract and in the discussion we add figures illustrating in graphite form the results of the analysis, summarizing the processes from Table 1 and the tables in the Results section (in the graphical abstract - a simpler presentation of the results, and in the Discussion - a large detailed and clear drawing).

After changes are made to the manuscript, it must be reviewed by a fluent English-speaking entomologist.

**Language Note:** The Academic Editor has identified that the English language must be improved. PeerJ can provide language editing services - please contact us at [email protected] for pricing (be sure to provide your manuscript number and title). Alternatively, you should make your own arrangements to improve the language quality and provide details in your response letter. – PeerJ Staff

·

Basic reporting

The use of biological methods to protect plants from pests contributes to the development of sustainable agriculture and forestry. Cultivation of harmful insect host species in laboratory conditions for the purpose of propagation of parasitoids is important for biological control. Therefore, the topic of the article under review is very relevant. The research questions and hypotheses are clearly defined in the manuscript. The methods are described, only some points require clarification. The conclusions are confirmed by statistical analysis. However, there are some shortcomings and technical comments.

At the first mention of the Latin name of an insect pest (Gonipterus platensis) and a plant (Eucalyptus urophylla) in the abstract and in the main text of the manuscript, it is necessary to indicate the author’s surname and the year of description of the species.

In the abstract there is frequent repetition of the Latin name of the species «G. platensis», twice in one sentence (24–25). I recommend rephrasing the sentence.

The text in the «Results» section is disproportionately small. I recommend moving some sentences describing the experimental part from the «Discussion» to the «Results».

In the «Discussion» section, you can add scientific information about the biology of related species of the genus Gonipterus. About 20 species of this genus have been described. And draw an analogy. The conditions for effective cultivation of these species may be known.

I recommend adding the word «polyandry» (44–45) to Keywords. In this article, this is the key word that carries the main semantic load.

The text of the manuscript is composed mainly of very long sentences that are difficult for the reader to understand. I recommend breaking your sentences into at least two parts.

Experimental design

It is not entirely clear how the mating of G. platensis females with different males was recorded in the polyandry variant. Was there constant surveillance? Videography? I recommend that you address this issue in the «Materials and Methods» section.

Repeated identical paragraphs of text in the «Materials and Methods» section (149–154, 172–177). One needs to be removed.

Validity of the findings

Some sentences lack punctuation marks (95, 171).

When citing scientific literature in the text, errors in the spelling of the names of the authors (285–286) are noted.

According to the citation rules, there must be an ampersand between the two names of the authors. In the text in the links between the names of the authors of publications, it is necessary to replace the union «and» with the graphic symbol «&». For example: 83, 89, 92, 101, 104, etc.

Quoting the same publication twice in one paragraph is not acceptable. One link must be removed from the text (Introduction 83, 85).

Additional comments

No comments yet.

Reviewer 2 ·

Basic reporting

no comment

Experimental design

no comment

Validity of the findings

no comment

Reviewer 3 ·

Basic reporting

no comment

Experimental design

no comment

Validity of the findings

no comment

Additional comments

Upon review, I noted several comments and recommendations (see attached pdf file) for manuscript improvement. Firstly, it is suggested to replace the term "monogamy" with "monandry" to accurately reflect the focus on female mate choice. Secondly, ensuring compliance with the ICZN Code for all species names is essential for scientific accuracy. Thirdly, merging certain paragraphs in the methods section, as suggested in the attached file, would enhance clarity and coherence.
Furthermore, addressing discrepancies in Table 2, where standard error is used while empirical probability is provided in other tables, is recommended for consistency. Simplifying and making the titles of all tables shorter, formal, and more informative would improve readability. Lastly, supplementing the results with graphs or other visualizations could enhance the manuscript's value by providing a clearer representation of the data.

Annotated reviews are not available for download in order to protect the identity of reviewers who chose to remain anonymous.

---

## Round 0.2 · accepted · Accept

The article is really interesting and practically important. Its results will make it possible to optimize the reproduction and maintenance in laboratory conditions of this important pest species for the development of biological and integrated methods of combating it.